# A Potential New Role for Zinc in Age-Related Macular Degeneration through Regulation of Endothelial Fenestration

**DOI:** 10.3390/ijms222111974

**Published:** 2021-11-05

**Authors:** Fiona Cunningham, Sabrina Cahyadi, Imre Lengyel

**Affiliations:** 1Wellcome-Wolfson Institute of Experimental Medicine, Queen’s University, Belfast BT9 7BL, UK; F.Cunningham@qub.ac.uk; 2UCL Institute of Ophthalmology, University College London, London EC1V 9EL, UK; sab.cahyadi@gmail.com

**Keywords:** zinc, fenestration, choroid, age-related macular degeneration, PV-1

## Abstract

Age-related macular degeneration (AMD) is a common blinding disease in the western world that is linked to the loss of fenestration in the choriocapillaris that sustains the retinal pigment epithelium and photoreceptors in the back of the eye. Changes in ocular and systemic zinc concentrations have been associated with AMD; therefore, we hypothesized that these changes might be directly involved in fenestrae formation. To test this hypothesis, an endothelial cell (bEND.5) model for fenestrae formation was treated with different concentrations of zinc sulfate (ZnSO_4_) solution for up to 20 h. Fenestrae were visualized by staining for Plasmalemmal Vesicle Associated Protein-1 (PV-1), the protein that forms the diaphragms of the fenestrated endothelium. Size and distribution were monitored by transmission electron microscopy (TEM). We found that zinc induced the redistribution of PV-1 into areas called sieve plates containing ~70-nm uniform size and typical morphology fenestrae. As AMD is associated with reduced zinc concentrations in the serum and in ocular tissues, and dietary zinc supplementation is recommended to slow disease progression, we propose here that the elevation of zinc concentration may restore choriocapillaris fenestration resulting in improved nutrient flow and clearance of waste material in the retina.

## 1. Introduction

Age-related macular degeneration (AMD) is a degenerative disease of the outer retina that leads to irreversible vision loss and legal blindness [1]. Pharmacological treatments for AMD delay but do not prevent disease progression, involve invasive procedures, and are costly [2]. Alternative approaches are increased dietary intake of zinc-rich food or supplementation of zinc with a cocktail of antioxidants that have been shown to delay AMD progression [3,4,5].

Zinc is a ubiquitous trace element present throughout the human body [6]. Many of the functions of zinc are enabled by its ability to interact with amino acids and binding to proteins [7]. One of the highest tissue concentrations of zinc in the human body is found in the retina/choroid complex in the eye [6]. The innermost layer of the choroid is a capillary network called the choriocapillaris. The choriocapillary endothelium enables efficient delivery of oxygen and nutrients and removal of metabolic waste into the peripheral circulation [8].

The high permeability of the choriocapillaris compared to other capillary beds is a result of its fenestrated phenotype [8]. Fenestrae are pores in the cell membrane formed by the fusion of the plasma membrane [9]. The size of fenestrae is dependent on the presence of a diaphragm formed by dimers of the protein plasmalemma vesicle-associated protein-1 (PV-1) [9]. Fibrils arrange radially within the pore, like spokes in a wheel, around a central knot [10]. Data extracted from in vitro models show that remodeling of the actin cytoskeleton is essential for fenestrae formation, while PV-1 is required for fenestrae morphogenesis [9].

In AMD, fenestrae are not observed in areas where the retinal pigment epithelium (RPE) has atrophied. The loss of signaling through RPE-derived vascular endothelial growth factor (VEGF) may be one factor contributing to this defenestration [11,12]. However, it is possible that other biomolecules may also contribute to fenestrae formation. Here, it was hypothesized that zinc, due to its high bioavailability in the outer retina and choroid, may be an additional factor regulating endothelial fenestration.

To test this hypothesis, an endothelial cell model for fenestrae formation (bEND.5) was treated with zinc sulfate (ZnSO_4_), and localization of PV-1 was assessed by immunofluorescent staining, and the sizes of fenestrae were determined by transmission electron microscopy (TEM).

## 2. Results

### 2.1. Measurement of Bioavailable Zinc in the Culture Media for bEND.5 Cells

In the present study, bEND.5 cells were maintained in DMEM supplemented with glutamine, pyruvate, 10% fetal calf serum (FCS), antibiotics, non-essential amino acids (NEAA), and β-mercaptoethanol (β-ME). Zinc supplementation was in the form of zinc sulfate (ZnSO_4_). As reported earlier, bioavailable zinc was below 100 nM even as high as 175 μM added ZnSO_4_ in bEND.5 cell culture media [7,13]. We assessed the toxicity of zinc using propidium iodide and calcein and did not find signs of cell damage in the concentration of added zinc we used (data not shown).

### 2.2. Fenestration of BEND.5 Cells

It has been shown that bEND.5 cells could develop fenestrae in vitro after incubation with 1.25 μM of latrunculin A (LA) or 200 ng/mL VEGF for 3 h [9]. We used these treatments as controls in our experiments (Figure 1A,B). Cells on glass coverslips were fixed and stained for PV-1 alongside the fluorescent actin cytoskeleton probe, rhodamine-phalloidin (Figure 1A,B). Cells treated with LA (Figure 1A) or VEGF (Figure 1B) had notable rearrangements of the actin cytoskeleton and redistribution of PV-1 into areas previously described as ‘sieve plates’ [9]. To confirm that these areas contain fenestrae, cells were grown on TEM grids, fixed, and imaged directly (Figure 1C). Rearrangement of the actin cytoskeleton into bundles could be observed at 5000× magnification. In between these bundles, we observed uniform-sized fenestrae. At higher magnification, details of fenestrae could be seen, including the central knot.

### 2.3. Zinc Induces a Concentration and Time-Dependent Rearrangement of PV-1 in bEND.5 Cells

To determine if zinc could induce a concentration-dependent rearrangement of PV-1, we added increasing concentrations of ZnSO_4_ for 20 h to the culture medium (Figure 2 and Figure 3). In control untreated cells (Figure 2A–D), PV-1 was homogenously distributed throughout the cytoplasm, with perinuclear enrichment. In cells treated with ≥100 μM of total ZnSO_4_ (Figure 2E–T), PV-1 is redistributed to form small clumps like those seen previously in the presence of LA or VEGF-A_165_ (Figure 1). There was no appreciable difference between 125 μM and 175 μM of added ZnSO_4_. However, at 200 μM, signs of cell toxicity were apparent, including loss of membrane integrity, cell shrinkage and condensation and breakup of the nucleus (Figure 2Q–T). Therefore, for further experiments, 125 μM of added ZnSO_4_ was used. Sieve plates induced by 125 μM of zinc appeared to be smaller in size compared to those induced by LA or VEGF-A_165_, and there was only mild actin cytoskeleton rearrangement based on phalloidin staining.

To determine the time needed for PV-1 rearrangement, bEND.5 cells were treated with 125 μM of ZnSO_4_ for 3, 6, 9, or 20 h. Cells were fixed and stained for PV-1 and with rhodamine-phalloidin (Figure 3). As early as 3 h (Figure 3E–H), redistribution of PV-1 into sieve plates could be observed, in comparison to the homogenous distribution (and intense perinuclear staining) of this protein in control cells (Figure 3A–D). Redistribution of PV-1 was more apparent after 6, 9, and 20 h of zinc treatment. The data indicate that zinc can change endothelial cell fenestration as early as 3 h and possibly earlier.

### 2.4. Size Distribution of Zinc-Induced Fenestrae Were Similar to Those Induced by LA

While translocation of PV-1 immunoreactivity is an indication, it is not a confirmation of endothelial fenestrae formation. Therefore, we investigated fenestrae formation and size distribution by TEM. bEND.5 cells were cultured on formvar-coated nickel TEM grids. Cells were treated with LA for 3 h or 125 μM of ZnSO_4_ for 20 h. Cells were fixed and viewed under a JEOL10TEM microscope (Figure 4). LA treatment induced the formation of sieve plates arranged between bundles of the actin cytoskeleton (Figure 4A), and within these sieve plates, multiple uniform fenestrae with central diaphragms could be observed at higher magnification (Figure 4B,C). Untreated bEND.5 cells showed no fenestration or formation of actin bundles (Figure 4D–F). Treatment with 125 μM of ZnSO_4_ induced smaller sieve plates arranged between small actin cytoskeleton bundles (Figure 4G). Fenestrae appeared uniform with central diaphragms at higher magnifications (Figure 4H,I).

The size of fenestrae was assessed using ImageJ from the TEM images taken at 12,000× magnification. We measured 393 fenestrae from the ZnSO_4_-treated and 775 fenestrae from LA-treated cells and their size distribution was plotted in Figure 5. Fenestrae of ZnSO_4_-treated cells had a mean diameter of 69 ± 25 nm, similar to the 72 ± 30 nm in the presence of LA. The data indicate that zinc alone can induce endothelial cell fenestration comparable to that of LA. This is further demonstrated in Table A1 and Figure A1 in Appendix A.

## 3. Discussion

Deficiencies of zinc have been associated with poor diet and aging. It has been suggested that zinc concentrations are reduced in the peripheral circulation and the outer retina in AMD [4,14]. In response to these observations, in clinical studies, zinc supplementation or zinc-enriched food have been shown to delay the progression of intermediate AMD to end-stage disease [4,5]. However, the biological explanation for the beneficial effects is not yet fully elucidated. It is believed that zinc affects oxidative stress, impaired autophagy, and lipofuscin accumulation in the RPE [14,15], but the effects on the choroidal endothelium are less well studied. A recent review suggested that zinc in the choroid may contribute to AMD via the perpetuation of choroidal inflammation [6]. In this study, we showed that elevated zinc could directly induce fenestrae formation in the choroidal microcapillary endothelium.

The murine brain endothelial cell line bEND.5 has been shown to develop fenestrae in vitro in response to LA and VEGF [9]. This was replicated in our study to pave the way for the experiments with elevated zinc concentrations. In our experiments, we supplemented zinc in the form of ZnSO_4_, a water-soluble salt that did not precipitate in the culture medium. While we added up to 200 μM of zinc to the culture medium, it is important to emphasize that the bioavailable zinc concentration remained below 100 nM due to the buffering capacity of the culture medium [7,13,16]. However, it was not established in this study if the zinc buffering capacity of the media could be saturated at higher concentrations >200 μM.

Like LA and VEGF, zinc induced the reorganization of PV-1 into the so-called sieve plates. The sieve plates were smaller in appearance, and the zinc treatment was not associated with the apparent rearrangement of the actin cytoskeleton in the presence of LA. A previous in vitro study has shown that actin disassembly is essential for fenestrae formation in bEND.5 cells, likely by bringing apical and basal membranes into close proximity for fusion [9]. While not as robust as with LA, TEM images showed that zinc also helped the rearrangement of the cytoskeleton in bEND.5 cells. Previous studies have shown that zinc has an effect on the actin cytoskeleton in a variety of cell types, and this might be the case in our experiments also [17,18]. Zinc might also interact with components of the cell culture medium. The serum used in this study may contain low concentrations of growth factors, such as VEGF, that may have a synergistic interaction with zinc to induce fenestrae formation. In control untreated cells, occasionally sporadic fenestrae could be observed, suggesting that factors in the serum enabled some spatial reorganization of the actin cytoskeleton. However, zinc might be required to induce the formation of sieve plates with proper fenestrae morphology by fostering the redistribution of PV-1.

In mice, soluble RPE-derived VEGF has been shown to play a role in maintaining choriocapillaris fenestration [12]. More recently, in an oxygen-induced retinopathy model, it has been reported that levels of VEGF-A were positively correlated with PV-1 levels, fenestration, and permeability in the choroid [19]. However, when PV-1 expression was inhibited by siRNA, fenestration was lost, even at high VEGF concentrations [19]. These suggest that PV-1 is the critical molecule for choriocapillaris fenestration. The role of zinc in fenestrae formation had not yet previously been investigated. Based on our results, we propose that in the presence of physiological zinc concentrations, fenestration could still occur in vivo even if VEGF levels were decreased. Most likely, in vivo, zinc may act as a co-factor to regulate the recruitment of PV-1 and the formation of fenestrae by working in synergy with VEGF. VEGF alone could maintain fenestration even at decreased zinc levels, although this might provoke unwanted side effects like choroidal neovascularization. Restoring zinc status might be behind the slowing progression to end-stage AMD, reported by the AREDS study [4]. Further experiments will be necessary to evaluate how the re-balancing of zinc and VEGF levels affect endothelial fenestration and/or neovascularization. Based on our experiences, studying the titration of zinc and VEGF in the bEND.5 model is highly variable. Progress on this will likely require experiments in vitro.

All in all, maintaining normal zinc levels in the peripheral circulation and local tissues is becoming essential for maintaining the outer blood-retinal barrier. This study on endothelial cells and previous works on epithelial cells show that maintaining sufficient zinc concentrations at the RPE/choroid interface is likely to be critical to maintaining healthy cellular functions and cell-to-cell communication and transport [7,13]. Both forms of end-stage AMD (neovascular AMD and geographic atrophy) are preceded by common pathological changes, namely sub-RPE deposit formation. Zinc alone, or in synergy with other molecules, may maintain choroidal fenestration and be beneficial in slowing the progression of AMD to end-stage disease by improving the removal of waste products from the RPE and the photoreceptor outer segments. These findings could help to understand clinical supplementation and nutritional studies and inform further zinc supplementation guidelines for AMD patients at different stages of the disease.

## 4. Materials and Methods

### 4.1. Cell Culture

The murine brain endothelial cell line bEnd.5 was obtained from the European Collection of Authenticated Cell Cultures (96091930, ECACC, UK). bEnd.5 were grown in high-glucose (4.5 g/L) DMEM (with glutamine and pyruvate) (11885084, Life Technologies, UK) containing 10% FCS (10500064, Life Technologies, UK), 1% penicillin-streptomycin (15070063, Life Technologies, UK), 1× non-essential amino acids (M7145, Merck, Germany), and 5 μM of β-mercaptoethanol (31350010, Life Technologies, UK) in cell culture flasks coated with 2 μg/cm^2^ of poly-l-lysine (P4707, Merck, Germany) and maintained in an incubator at 37 °C and 5% CO_2_.

Twenty-four hours before fenestrae induction, bEnd.5 were seeded on glass coverslips or transmission electron microscopy (TEM) grids (Agar Scientific, England, UK) in complete cell culture media. Glass coverslips were coated with 1% bovine gelatin (G1393, Merck, Germany) in 24-well plates. bEnd.5 were seeded at a density of 3.3 × 10^4^ cells per well. Formvar-coated nickel TEM grids were coated with 1% bovine gelatin before seeding 2.5 × 10^4^ cells per grid. Cells were left overnight to facilitate attachment.

### 4.2. Fenestrae Induction

Cells were treated with either 1.25 μM of latrunculin A (LA) (L12370, Invitrogen, MA, USA) for 3 h, increasing concentrations (50–200 ng/mL) of VEGF-A_165_ (450-32, Peprotech, UK) for 4 h, or increasing concentrations (100–200 μM) of zinc sulfate (ZnSO_4_) (Z2876, Merck, Germany) for up to 20 h. Cells were incubated at 37 °C with 5% CO_2_ for the duration of the treatment.

### 4.3. Immunocytochemistry

bEnd.5 treated on coverslips were fixed with 4% PFA (methanol free) (28906, Thermo Fisher Scientific, UK) for 10 min and washed with 1× PBS. Prior to immunofluorescent staining, cells were permeabilized with 0.1% Triton X-100 (X100, Merck KGaA, Darmstadt, Germany) diluted in PBS (10010015, Life Technologies, UK) for 15 min. Rat anti-mouse PV-1 antibody (gifted by Dr. David Shima) was diluted 1:400 in a blocking solution composed of 0.2% fish skin gelatin (G7765, Merck KGaA, Darmstadt, Germany) and 5% normal goat serum (G9023, Merck, Germany) in PBS. Cells were incubated with primary antibody for 60 min before incubation with rhodamine-phalloidin (1:300) (R415, Invitrogen, MA, USA) and goat anti-rat Alexa Fluor 488 (1:1000) (A-11006, Invitrogen, MA, USA) for a further 60 min at room temp. Cells were rinsed and mounted on glass microscope slides with VectaShield mounting medium containing DAPI (H-1200-10, Vector Laboratories, Burlingame, CA, USA). Images were captured using the LSM700 confocal microscope (Carl Zeiss Microscopy GmbH, Jena, Germany) at 60× magnification.

### 4.4. Transmission Electron Microscopy (TEM)

Cells treated on TEM grids were fixed in Karnovsky’s solution overnight at 4 °C and washed with 1× PBS. bEnd.5 were then post-fixed in 1% osmium tetroxide (75633, Sigma-Aldrich, UK) for 30 min and washed with distilled water. Following fixation, cells were placed through a series of increasing ethanol (E/0665DF/17, Thermo Fisher Scientific, UK) concentrations (30–100%) and finally incubated in hexamethyldisilazane (440191, sigma-Aldrich, UK) before drying overnight. Cells were viewed at high magnification using a JEOL10TEM microscope.

### 4.5. Measurement of Fenestrae Diameters

TEM images of LA or ZnSO_4_-treated bEnd.5 were acquired at 12,000× magnification and imported into ImageJ (NIH, Bethesda, MD, USA). Images were despeckled and outliers removed. The areas and diameters of individual fenestrae were determined using the analyze particles feature. Objects < 40 nm in diameter were rejected from the analysis based on a manual assessment that determined the smallest fenestrae to be 46 nm in diameter. A total of 775 fenestrae from LA-treated cells and 393 fenestrae from ZnSO_4_-treated cells were assessed from 3 independent experiments.

## Figures and Tables

**Figure 1 ijms-22-11974-f001:**
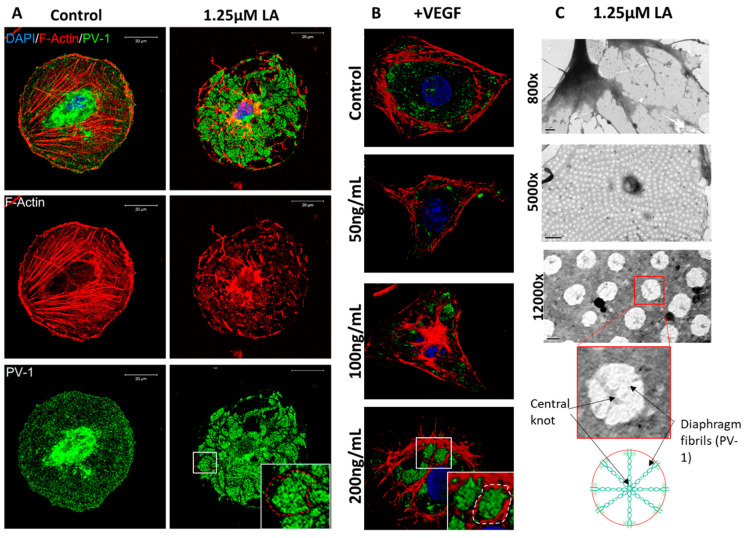
bEND.5 cells have the potential to become fenestrated. (**A**). BEND.5 cells treated with 1.25 μM of LA had PV-1 (green) and actin (red) redistribution to sieve plates as compared to control non-treated cells (sieve plate indicated by white box and red dotted line). (**B**). VEGF-induced sieve plate formation could also be stimulated in bEND.5 cells as illustrated by the rearrangement of PV-1 (green) and the actin cytoskeleton (red) (white box and dotted line indicate sieve plate formation in 200 ng/mL of VEGF-treated bEND.5 cells). (**C**). TEM showed uniform distribution of fenestrae (800× magnification, scale bar: 1 µM; 5000× magnification, scale bar: 0.5 µM). At high magnification, fenestrae diaphragm fibrils and central knots could be visualized (12,000× magnification, scale bar: 200 nm) as highlighted in the accompanying diagram.

**Figure 2 ijms-22-11974-f002:**
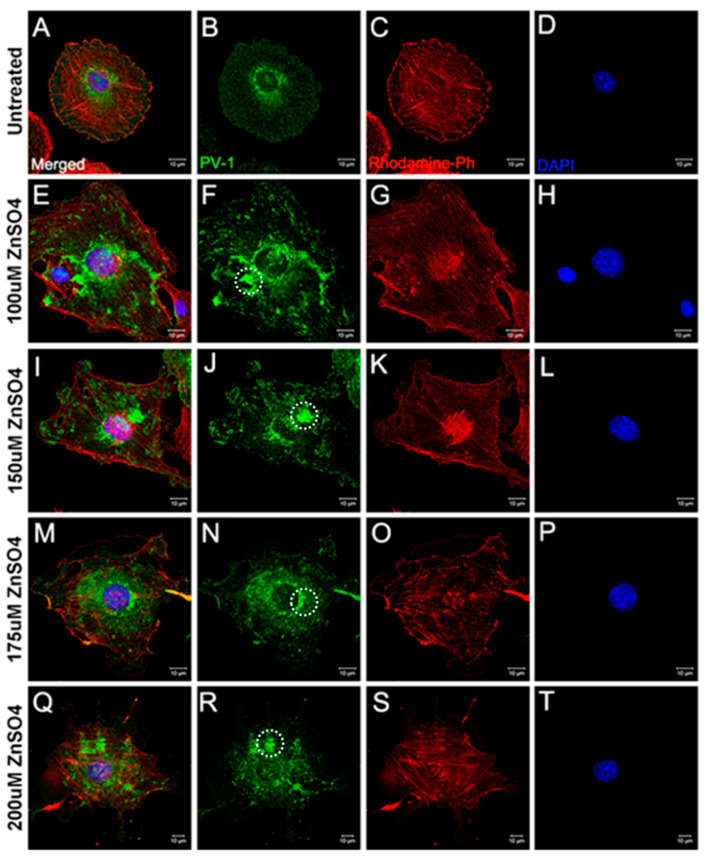
Sieve plate formation induced by zinc sulfate. 100–200 μM of ZnSO_4_ induced PV-1 redistribution (green) and sieve plate formation (**F**,**J**,**N**,**R**), examples indicated by white dotted circles) compared to untreated control cells (**B**). However, there were no significant actin (red) cytoskeletal changes (**C**,**G**,**K**,**O**,**S**) or degradation of the nucleus (blue) (**D**,**H**,**L**,**P**,**T**). (**A**,**E**,**I**,**M**,**Q**) show the images after merging the different fluorescence channels.

**Figure 3 ijms-22-11974-f003:**
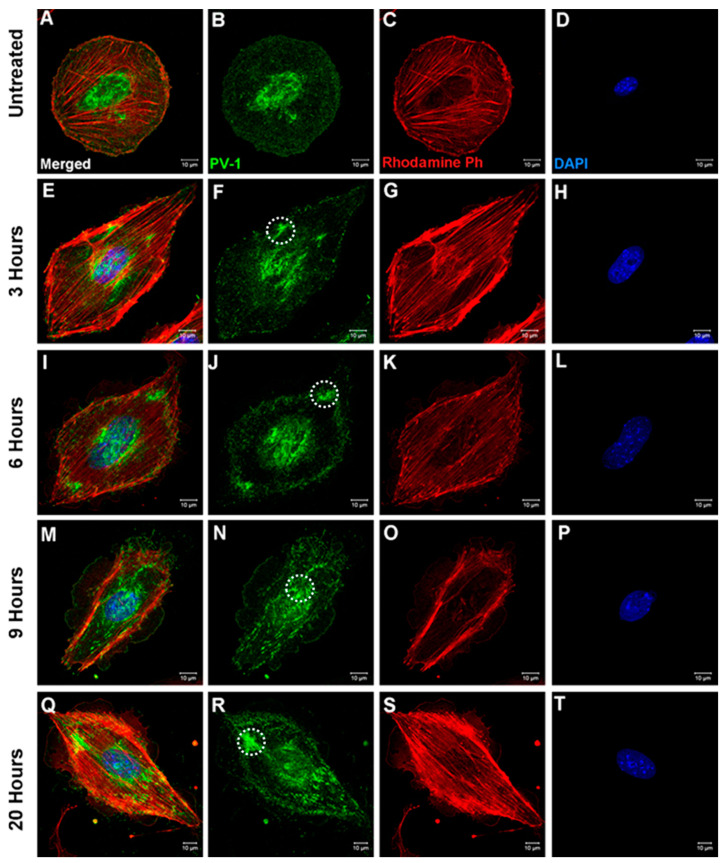
Time-dependent sieve plate formation induced by zinc. When compared to control (**B**) 125 μM ZnSO_4_ induced PV-1 (green) rearrangement as early as 3 h after treatment initiation (**F**); however, sieve plate formation became clearly apparent in cells treated for 9 or more hours (**N**,**R**). Sieve plate formation is indicated by white dotted circles (**F**,**J**,**N**,**R**). There were no significant actin (red) cytoskeletal changes (**C**,**G**,**K**,**O**,**S**) or degradation of the nucleus (blue) (**D**,**H**,**L**,**P**,**T**). (**A**,**E**,**I**,**M**,**Q**) show the images after merging the different fluorescence channels.

**Figure 4 ijms-22-11974-f004:**
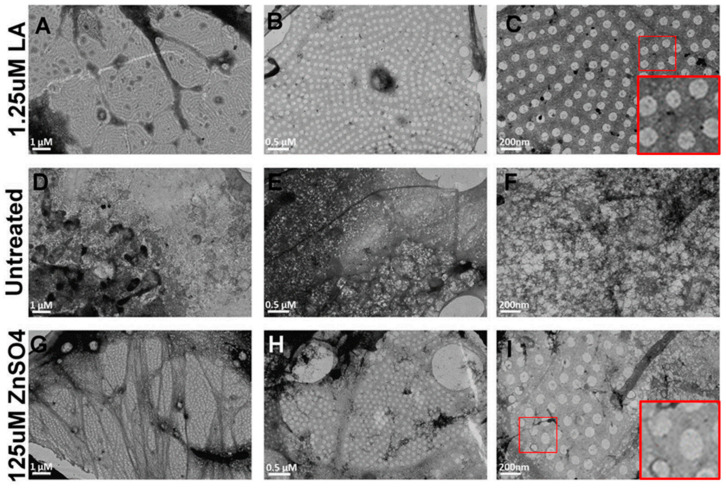
Zinc-induced fenestrae formation was examined by TEM. Fenestrae formation induced by LA (**A**–**C**) was used as a positive control for these observations. In untreated cells, there was some spontaneous fenestrae formation; however, these lacked uniformity and diaphragm organization (**D**–**F**). As compared to LA-treated cells, zinc induced comparatively smaller sieve plates without major cytoskeletal rearrangements (**G**,**H**). However, zinc-induced fenestrae were uniform and exhibited typical morphology, including diaphragm formation (**I**), like those induced by LA (**C**). (Scale bar, (**A**,**D**,**G**) = 1 micron; (**B**,**E**,**H**) = 0.5 micron; (**C**,**F**,**H**) = 200 nm).

**Figure 5 ijms-22-11974-f005:**
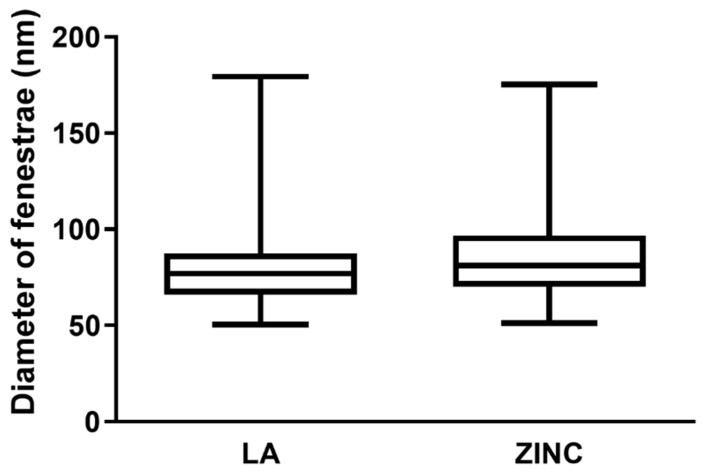
Size distribution of fenestrae diameter induced by zinc or LA. The distribution of fenestrae diameters was similar in LA and ZnSO_4_ treated cells. A total of 775 fenestrae from LA-treated cells and 393 fenestrae from ZnSO_4_-treated cells were assessed from 3 independent experiments.

## Data Availability

Not applicable.

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
