# Peer review of "A Potential New Role for Zinc in Age-Related Macular Degeneration through Regulation of Endothelial Fenestration"

_ijms, 2021, doi:10.3390/ijms222111974_

Round 1

Reviewer 1 Report

Summary:
The authors have shown that treatment of bEND.5 cells with ZnSO4 can cause a rearrangement of PV-1 protein and actin-cytoskeleton consistent with the formation of fenestrae, suggesting that zinc is necessary for the formation of these pits. This may have wider implications into AMD as fenestrae are not noted in areas of RPE depletion, and serum Zinc has been noted to be low in AMD patients.

Review:
The concepts presented in this paper are suitably interesting and ask some relevant questions that can be expanded upon regarding zinc supplementation in AMD patients. I have some issues with the presentation of the figures and data and ask the reviewers to better clarify their images to assist readers not well versed in IF/EM to understand. Additionally, to expand I suggest a couple other experiments.

Major Suggestions:
1) Have the authors considered performing the Zn supplementation in tandem with the LA or VEGF experiments to see if this enhances the fenestrae formation or size? This might suggest a relationship between zinc metabolism and the VEGF/LA pathways
2) I presume based on the discussion of AMD in the same breath as VEGF, the authors are suggesting this a primary role in neovascular/’wet’ AMD? Or does this have implications in dry AMD as well?

Minor Suggestions:
1) IF images, Particularly in Figure 1 I would request that the authors plainly label what is a control, and what is an experimental image within each figure as I’m having a hard time distinguishing based on their description. In Figure 1, I do not believe there is a control image, which would clarify the effects of LA/VEGF. Figure 2 is arranged well; I’d appreciate if Figure 1 could be arranged in this manner.
2) The authors claim the formation of ‘sieve plates’ and other structures in each of these figures, I’d request the authors plainly indicate what structures are which using arrows or other indicators. This goes for all figures. Not all readers will be well versed in the interpretation of IF/EM images and this will help clarify what exactly is being referred to. I’d also request they adjust the arrows on the EM pictures in Figure 1J to better indicate what is being referred to, or possibly provide a different image as I’m unable to tell what the authors are referring to in this image
3) More a question out of curiosity, why did the authors choose ZnSO4 when another salt such as ZnCl2 has a much higher solubility in water?

Author Response

1) Have the authors considered performing the Zn supplementation in tandem with the LA or VEGF experiments to see if this enhances the fenestrae formation or size? This might suggest a relationship between zinc metabolism and the VEGF/LA pathways

We agree with the Reviewers titration of zinc and VEGF is an exciting question. Titrating ‘out’ VEGF while titrating zinc ‘in’ to determine the concentrations of zinc that might enhance VEGF-induced fenestration is an experiment we have tried, and that is continuously on our radar. However, we found that using  bEND.5 cells for these experiments is highly variable and unsuitable for this work. bEND.5 is excellent for looking at the factors that influence fenestration, though. This is why we chose to report our result with zinc on these cells. This has now been more clearly addressed in the paragraph between lines 249-266.

2) I presume based on the discussion of AMD in the same breath as VEGF, the authors are suggesting this a primary role in neovascular/’wet’ AMD? Or does this have implications in dry AMD as well?

Thank you for this question. We believe zinc affects both neovascular and dry AMD. Altered zinc milieu probably precedes end-stage and is likely to start at early or intermediate stages. These include Bruch’s membrane thickening and sub-RPE deposit formation.  Reduced choriocapillaris fenestration could contribute RPE changes by compromising waste clearance in the outer retina. Zinc supplementation may have positive effects in reducing the progression of AMD to its vision-threatening end stages by ‘reopening’ fenestrae in the choriocapillaris to provide better clearance of waste. As we have reported previously that the same zinc treatment has many effects on the RPE as well. Updates have been made in the manuscript discussion to clarify this (lines 269-273).

Minor suggestions:

1) IF images, Particularly in Figure 1 I would request that the authors plainly label what is a control, and what is an experimental image within each figure as I’m having a hard time distinguishing based on their description. In Figure 1, I do not believe there is a control image, which would clarify the effects of LA/VEGF. Figure 2 is arranged well; I’d appreciate if Figure 1 could be arranged in this manner.

We want to thank the Reviewer for asking for clarification. Control and experimental images have been more distinctly labelled now. Initially, there was no control image provided for the LA experiment panels. This has now been updated. The layout of figure 1 has been updated accordingly. The figure legend (lines 83-89) have been updated to reflect this.

2) The authors claim the formation of ‘sieve plates’ and other structures in each of these figures, I’d request the authors plainly indicate what structures are which using arrows or other indicators. This goes for all figures. Not all readers will be well versed in the interpretation of IF/EM images and this will help clarify what exactly is being referred to. I’d also request they adjust the arrows on the EM pictures in Figure 1J to better indicate what is being referred to, or possibly provide a different image as I’m unable to tell what the authors are referring to in this image

Again, we want to thank the Reviewer for helping to improve the understanding of the images. In the new figure 1A on page 13, a sieve plate has been indicated with a box, and dotted line with a zoomed-in image of the structure. This has been repeated in the new Figure 1B. In the new Figure 1C, to aid the description of the central knot and diaphragm fibril structures, a larger image of a pore is provided with an accompanying schematic for clarity. Updates have been made to the figure legend accordingly. In figures 2 and 3, circles indicate examples of sieve plates in images of PV-1 staining. The figure legends have been updated accordingly. In figure 4, larger images of the fenestrae are provided in panels C and I.

3) More a question out of curiosity, why did the authors choose ZnSO4 when another salt such as ZnCl2 has a much higher solubility in water?

The Reviewer is raising an interesting question. It would indeed be fascinating to determine whether one or another supplement works the best. Historically, we built a significant experience on using ZnSO4. That is why we used this in our experiments.

Reviewer 2 Report

The manuscript describes the endothelial fenestration induced by Zn in an endothelial cell model. The manuscript contains a very substantial body of experiments that are well-conducted and in general well described. 

The study is accurate and detailed, and it represents a significant addition to the research line carried out by this research group. I have only a couple of comments:

-Have the authors performed a toxicity assay for ZnSO4 in these cells? 125µM ZnSO4 could be toxic for cells.

-Have the authors performed Zn treatment and LA or VEGF at the same time in order to see synergistic effects on the fenestration formation?

-Have the authors considered studying the Zn effect on the fenestration process in an animal model?

- I suggest including labels as “control” or “LA” or “VEGF” treatment in the panels of figure 1b.

-I would like to see in figure 2, figure 3 that green corresponds with PV-1 staining and the same for the rest of the colours.

-I would like to see in figure5 the number of experiments represented.

Author Response

1) Have the authors performed a toxicity assay for ZnSO4 in these cells? 125µM ZnSO4 could be toxic for cells.

Thank you for highlighting the lack of this vital information. We used propidium iodide and calcein and found that the ZnSO4 treatment used in these experiments were not toxic to the cells. We inserted a sentence at Line 63 to state this.

2) Have the authors performed Zn treatment and LA or VEGF at the same time in order to see synergistic effects on the fenestration formation?

We agree with the Reviewers titration of zinc and VEGF is an exciting question. Titrating ‘out’ VEGF while titrating zinc ‘in’ to determine the concentrations of zinc that might enhance VEGF-induced fenestration is an experiment we have tried, and that is continuously on our radar. However, we found that using  bEND.5 cells for these experiments is highly variable and unsuitable for this work. bEND.5 is excellent for looking at the factors that influence fenestration, though. This is why we chose to report our result with zinc on these cells. This has now been more clearly addressed in the paragraph between lines 249-266.

3) Have the authors considered studying the Zn effect on the fenestration process in an animal model?

Thank you for this question. It is indeed a significant issue. We are in the process to supplement animal models with zinc, but the data is not yet available. It is essential to mention that fenestration analysis in vivo is highly challenging as visualisation of fenestration on sections may not represent what happens in the entire back of the eye. Still, we hope we will be able to report at some point on our findings.

4) I suggest including labels as “control” or “LA” or “VEGF” treatment in the panels of figure 1b.

Thank you for the excellent suggestion. This figure has now been updated and restructured as per the comments from both reviewers. This also includes the addition of more evident labels.

5) I would like to see in figure 2, figure 3, that green corresponds with PV-1 staining and the same for the rest of the colours.

Thank you for asking for clarification on this. We have made sure that this information is accurately included in all text and figures and figure legends.

6) I would like to see in figure5 the number of experiments represented.

Thank you for this suggestion. The number of experiments is now included, and the text has been updated in the figure legend (lines 210 and 211).

Round 2

Reviewer 1 Report

I have reviewed the manuscript and I believe the authors have addressed all the issues raised by myself and the second reviewer. Thank you to the authors for the opportunity to review this manuscript and I wish them well on their future endeavours.